# Nutrition and Physical Activity as Modulators of Osteosarcopenic Adiposity: A Scoping Review and Recommendations for Future Research

**DOI:** 10.3390/nu15071619

**Published:** 2023-03-27

**Authors:** Vesna Vucic, Danijela Ristic-Medic, Aleksandra Arsic, Snjezana Petrovic, Marija Paunovic, Nadja Vasiljevic, Jasminka Z. Ilich

**Affiliations:** 1Group for Nutritional Biochemistry and Dietology, Centre of Research Excellence in Nutrition and Metabolism, Institute for Medical Research, National Institute of Republic of Serbia, University of Belgrade, 11000 Belgrade, Serbia; 2Institute of Hygiene and Medical Ecology, Medical Faculty University of Belgrade, 11000 Belgrade, Serbia; 3Institute for Successful Longevity, Florida State University, Tallahassee, FL 32306, USA

**Keywords:** osteosarcopenic adiposity, osteosarcopenic obesity, nutrients, nutritional status, physical activity, resistance training

## Abstract

Osteosarcopenic adiposity (OSA) syndrome denotes the confluence of bone, muscle, and adipose tissue deterioration. Being a complex entity, numerous uncertainties about OSA still exist, despite the extensive research on the topic. Our objectives were to evaluate human studies addressing dietary intake/nutritional status and the quantity/types of physical activity related to OSA. The search in PubMed, Scopus, and Web of Science databases was conducted to examine relevant articles published from inception to the end of December 2022, utilizing the MeSH strings in the search strategy. Only studies published in English and conducted in humans (≥18 years) without chronic conditions (cancers, kidney/liver disease) or pregnancy were used. Book chapters, abstracts-only, and studies in which participants did not have all three body composition components measured to identify OSA or when body composition components could not be related to the independent/exposure variables were excluded. A total of *n* = 1020 articles were retrieved from all three databases and eight more from the reference lists. After the exclusion of duplicates and other unsuitable articles, *n* = 23 studies were evaluated. Among those, eleven were from epidemiological or cross-sectional studies relating nutrients/dietary intake or nutritional status with OSA. Another four examined the relationship between serum biomarkers (vitamin D and ferritin) with OSA, while eight articles presented the results of the interventional studies with resistance training. Overall, higher protein, calcium, potassium, and vitamins D and C intakes emerged as nutrients positively modifying OSA, along with a diet higher in fruits and low-fat dairy foods. Higher serum vitamin D and ferritin were respectively positively and negatively related to OSA. Resistance training was a safe intervention yielding several beneficial outcomes for the OSA syndrome in older women.

## 1. Introduction

Osteosarcopenic adiposity (OSA) syndrome, originally coined as osteosarcopenic obesity (OSO), is the most advanced stage on the spectrum of body composition disorders. Its first identification and proof of concept were established in 2014 [1]. Briefly, OSA is a condition with simultaneous deterioration of bone (osteopenia/osteoporosis) and muscle (sarcopenia/dynapenia) with increased presence of body fat (adipose tissue). Body fat or adiposity may be manifested as an apparent overweight/obesity, or as a redistributed fat around organ tissues (visceral) and/or as an infiltrated fat (ectopic) into bone, muscle, and other organs [2]. The original term osteosarcopenic obesity was adjusted to osteosarcopenic adiposity [2] to better reflect the heterogeneity of adipose tissue: subcutaneous, visceral, and ectopic. Each produces different molecules that might render positive or negative health consequences. In the context of OSA, it is the excess of adiposity that leads to physiological and endocrine disturbances. Additionally, obesity, in general terms, typically refers to being overweight, most often defined by high body mass index (BMI), despite that the inadequacies of BMI for characterizing the body composition phenotypes and its paradoxical relations to morbidity/mortality have been widely recognized [2,3,4,5].

OSA concept was substantiated based on several established constructs characteristic for each of the three body composition tissues (bone, muscle, fat) [1,2], including (1) common precursors in the mesenchymal stem cells (osteoblasts, myocytes, adipocytes) and their deregulation leading to osteopenia/osteoporosis, sarcopenia/dynapenia, and compromised adipose tissue; (2) hormonal interactions among three tissues (e.g., osteocalcin and sclerostin released from bone cells; myostatins and troponins released from muscle cells; estrogens and adipokines released from adipocytes); (3) common etiologies of the impairment of each tissue; from lifestyle (overnutrition/malnutrition, lack of physical activity), to low-grade chronic inflammation, to changes in hormonal levels, to neuromuscular dysfunction; (4) aging and all accompanied inevitable changes; (5) easily observed deterioration in body composition phenotypes with the ensuing medical diagnoses [1,2,6,7,8,9]. Figure 1 depicts the OSA/OSO conceptualization as a body tissue triad.

Since its conception, OSA has been studied across the world in diverse populations and with different methods/techniques and cutoff criteria used for its identifications. As a result, no unique methodology has been identified, and a wide range of prevalence rates has been reported. So far, it has been associated with functional disabilities, increased frailty and risk of falls, systemic metabolic deregulation, and multiple chronic conditions [2,7]. Additionally, based on the characteristics of the syndrome, OSA may be associated with poor overall nutrition or inadequate intake of certain nutrients, as well as with a low engagement in physical activities of any kind [10,11]. However, no in-depth evaluation of such studies has been performed other than a small meta-analysis analyzing four studies that examined resistance training intervention effects on body composition and physical function in elderly OSA participants [12].

Keeping in mind the complex phenotype of OSA syndrome, we recommended earlier the nutritional and physical activity approaches for each of the conditions, as well as for the whole syndrome [7,8,10,11,13]. We also provided the algorithm for the nutritional and exercise treatment principles [7]. These were based on the already existing recommendations for each individual condition: osteoporosis, sarcopenia, and obesity. Namely, among nutrients, the intake of calcium, magnesium, vitamin D, and fiber, fitting the official recommendations [14], was enforced, while a higher-than-recommended intake of protein and omega-3 polyunsaturated fatty acids was suggested [13]. Regarding physical activity, we recommended a comprehensive exercise program to include weight bearing, aerobic, and strength training of moderate intensity for about 30–60 min 5 times/week [13]. All these recommendations were based on theoretical assumptions and scientific evidence extrapolated from other conditions. Therefore, it was necessary to perform a literature search and evaluate the original studies relating nutrition and physical activity with OSA syndrome. This will subsequently enable medical professionals to provide firsthand recommendations and management for OSA grounded on evidence-based data and clinical research.

In view of the above and because of the numerous uncertainties about OSA syndrome, the specific objectives of this scoping review were to evaluate the studies in human subjects addressing dietary intake and nutritional status, as well as the quantity and type of physical activity related to the syndrome. Additionally, we noted the prevalence of OSA, addressed the lack of common diagnostic criteria, and identified other gaps in current research, as well as highlighted the areas requiring further research. Nonetheless, the overall goal was to stimulate the launch of interventional clinical trials (currently lacking) that could lead to a better understanding of the processes within, as well as uncover the prevention and management strategies for OSA. The OSA and OSO terms and abbreviations (as well as osteopenic obesity/adiposity and sarcopenic obesity/adiposity) are used here interchangeably and follow those used in the original papers discussed.

## 2. Methods

### 2.1. Search Strategy

The search in PubMed, Scopus, and Web of Science databases was conducted to examine relevant articles published from inception to the end of December 2022. In the search strategies (strings adapted when necessary to fit the specific search requirements of each database), the following keywords and terms were used: Medical Subject Headings (MeSH) terms were used in the search strategy: ALL FIELDS osteoporotic OR osteosarcopenia OR osteopenia OR osteoporosis AND sarcopenia OR sarcopenic OR dynapenia AND obesity OR adiposity AND body fat OR bone mineral density, OR ALL FIELDS osteosarcopenic obesity OR osteosarcopenic adiposity, AND ALL FIELDS fractures OR fracture risk, AND ALL FIELDS nutrition OR nutrients OR physical activity, AND ALL FIELDS adults OR elderly OR postmenopausal women, NOT ALL FIELDS animal. Additionally, in order not to miss relevant articles, the reference lists of the related articles and reviews were examined, as well as the relevant articles found to be in-press or just published (January 2023) after the search was completed.

### 2.2. Eligibility Criteria

Only studies conducted in humans (≥18 years) and published in the English language were used, and no time limitation was applied. Based on the above criteria, co-authors screened the titles and abstracts and selected the eligible articles. Subsequently, potentially eligible full-text articles were downloaded and, after duplicates were removed, extracted and reviewed independently by the five co-authors (V.V., D.R.-M., A.A., S.P., and M.P.) with 10% of double checking. Book chapters, conference abstracts or abstracts-only, letters, articles with unusable information, and those conducted in children and adolescents, pregnant women, as well as those in animal models and cell cultures were excluded, as well as the studies in populations having chronic conditions (e.g., cancers, kidney/liver disease, HIV, or thyroid, glucocorticoid medication use). In addition, studies in which participants did not have all three body composition components (bone, muscle/lean, and fat tissues) measured to identify OSA or when all three body composition components could not be related to the independent/exposure variables (nutrition and physical activity), or when participants had only osteoporosis, or sarcopenia, or obesity, were excluded as well. The final set of articles was selected (by J.Z.I.) based on the following inclusion criteria: (1) observational and/or epidemiological studies; (2) clinical trials with nutrition and or physical activity intervention; (3) studies that considered nutrition, nutritional status, and/or physical activity as independent/exposure variables; (4) studies that included nutritional and/or physical activity biomarkers in blood/urine as independent/exposure variables. Any disagreement was settled by consensus among all authors.

### 2.3. Data Extraction

The following information was extracted from the included articles and presented in the tables: Name of the first author, publication year, search topic; Country and setting; Study design and duration of follow-up or intervention when applicable; Diagnostic criteria for each condition (osteopenia/osteoporosis, sarcopenia/dynapenia, adiposity); Sample size and sex of participants; Age range or mean age at baseline and follow-up, and/or mean age of participants with OSA/OSO; Prevalence/incident cases of OSA/OSO; Assessment tools used for identification of OSA/OSO, dietary intake or physical activity and/or lab analyses; Comparison of OSA/OSO participants with those having one or more body composition impairments (e.g., osteopenia/osteoporosis, osteopenic obesity osteosarcopenia, sarcopenia, sarcopenic obesity, obesity/adiposity alone) or normal.

## 3. Results

A total of *n* = 1020 articles were retrieved in the initial search from all three databases and eight more from the reference lists. After the exclusion of duplicates (*n* = 81), a total of *n* = 947 articles were screened, and *n* = 187 were fully assessed. Of those, *n* = 166 were excluded for various reasons (mostly for not having reported the values of all three-body composition tissues to identify OSA or inability to relate independent variables to the outcomes). Two papers published after the search was completed were also included in this review. Finally, *n* = 23 articles were included and evaluated in this scoping review. Figure 2 presents the flowchart (PRISMA) of the study selection process. Of note is that all studies are relatively new; the earliest one is from 2017 [15]. This is understandable, considering that OSA/OSO was identified just some 10 years ago, and the proof of concept was published in 2014 [1]. In the years prior, each of the body composition compartments was studied separately, or at the most, as a combination of two impaired tissues (most often sarcopenia and adiposity, as sarcopenic obesity) [1].

### 3.1. Dietary Intake, Nutritional Status, and OSA

Appendix A presents the characteristics and results of studies with dietary intake or nutritional status as independent/exposure variables related to OSA/OSO. Of the eleven studies in this section, eight were either cross-sectional or observational, examining the relationship between OSA and the nutritional status of the participants or their overall dietary intake and/or intake of specific nutrients [15,16,17,18,19,20,21,22]. Three of them also evaluated engagement in physical activity or a sedentary lifestyle [18,19,21]. In addition, one study was the 6-month randomized clinical trial investigating the effects of weight loss and dairy foods and/or calcium/vitamin D supplements on body composition [23], followed by the subsequent secondary analysis evaluating the cardiometabolic risk factors as the outcomes of the same intervention [24]. The only prospective study followed the participants for 5 and 10 years, examining the influence of energy-adjusted dietary inflammatory index (E-DII) on body composition and fracture rates [25]. The total number of evaluated participants was *n* = 19,151 (at baseline), with *n* = 12,143 (63.4%) women. The average age of the participants was 66.3 years, ranging from 50 to 95.2 years, with women being slightly older than men. The highest prevalence of OSA/OSO (when reported separately in women and men) was 91.9% in women [19] and 53.3% in men [17]. The lowest prevalence in women and men (reported as combined) was 6.4% [21]. The studies were conducted on five continents: Europe (Croatia) [16,17], the U.S. [23,24], Asia (South Korea) [15,18,19,20,22], South America (Brazil) [21], and Australia [25]. See Appendix A.

The important findings from these studies are described below and are listed in Appendix A. Briefly, Cvijetic et al.’s [16] study was conducted in several nursing homes, revealing that more than 1/3 of participants were at risk of developing malnutrition and ~6% were already malnourished (based on the Mini Nutritional Assessment), but there was no difference in the risk of malnutrition or malnourishment between those with or without OSA. This study was conducted during the COVID-19 pandemic and also revealed no difference in OSA prevalence or nutritional status in those who had COVID and those who had not to have a disease. However, a lower phase angle (indicating lower cell integrity and muscle quality) and a lower bone mass, while a higher intramuscular adipose tissue, were all significantly associated with the presence of OSA. Another, although smaller, study conducted in a nursing home utilized 24-hour dietary recalls for the assessment of participants’ dietary intake. The results showed a lower intake of protein, omega-3 polyunsaturated fatty acids, fiber, calcium, magnesium, potassium, and vitamins D and K—all below both European and U.S. recommendations [14,26] in all participants. However, those with OSA had significantly higher extracellular water, indicating a heightened inflammatory state in that group.

A six-month randomized clinical trial of weight loss employing a 25% reduction in energy intake complemented with either 4–5 servings of low-fat dairy foods or calcium-plus-vitamin D supplements (1500 mg/day and 600 IU/day, respectively) or placebo (control group) revealed improvement in all body composition components [23]. All participants lost weight, but those in the dairy group experienced higher loss in fat and a smaller loss in lean mass compared to the control group or the group taking supplements. The group with calcium/vitamin D supplements showed the best improvements in BMD in several skeletal sites compared to dairy or control groups [23]. Additionally, the subsequent secondary analysis in these participants (same 6-month intervention) showed improvement in blood pressure and numerous other cardiometabolic risk factors with weight loss in all participants, but significantly better in dairy and/or calcium/vitamin D supplements groups [24] (Appendix A). Since the evaluation of cardiometabolic outcomes was not a goal of this review, they are not discussed further but just noted as possible additional benefits to body composition and/or OSA.

The only prospective study in this section followed the participant for 5 and 10 years, investigating the influence of E-DII on body composition outcomes in a population of Australian women and men (*n* = 1098 at baseline) [25]. The results revealed that the consumption of a pro-inflammatory diet (higher E-DII scores) increased the incidence of fractures over 10 years in men but not in women, despite being associated with reductions in the lumbar spine and total hip BMD in both sexes. In addition, higher E-DII scores were significantly associated with higher fall risk and lower appendicular lean mass in men but not in women.

Each of the following five studies [15,18,19,20,22] utilized the Korea National Health and Nutrition Examination Survey (KNHANES) databases (from different years) to assess the relationship between several nutrients or foods [18,19,20] or Dietary Inflammatory Index (DII) [22], or Diet Quality Index International (DQI-I) [15] with OSA/OSO.

Among these, three studies [18,19,20] were published by the same group of researchers who used the KNHANES from 2008–2011, 2008–2009, and 2008–2010, respectively, thus having the advantage of large sample size (see Appendix A). Choi et al., 2021 [18] reported that lower intake of calcium was significantly associated with both osteosarcopenia and OSA, while physical activity was the lowest among the participants with OSA. In Choi et al.’s study [19], protein intake below the Korean recommendation (~0.9 g/kg body weight) was associated with higher odds of developing OSO in men. Additionally, intake of plant-based protein in men with OSO was higher than in men without (possibly indicating lower protein quality), while physical activity was significantly lower in men with OSO. No significant associations were noted in women, of whom 91.9% were identified as having OSO. Bae et al. [20] examined the intake of fruits, particularly those rich in vitamin C and potassium, in women only. They found that a lower intake of potassium and vitamin C and/or a lower intake of fruits was significantly associated with OSO. Kim et al. and Park et al. [15,22] used KHANES from 2009–2011 and 2008–2010, respectively. The latter found that the higher DII scores (denoting a higher proinflammatory diet) were significantly associated with the OSO phenotype [22], while the former study examining the DQI-I (higher scores denote better food quality), reported a significantly better body composition with higher DQI-I scores [15]. A small study conducted in Brazilian community dwelling women and men reported a significant association of lower protein intake (g/kg/weight, but not as a percentage of energy) in participants with OSO (women and men combined), while none of the other nutrients were significantly different among groups. The participants with OSO also had significantly lower grip strength and preferred a more sedentary lifestyle [21].

### 3.2. Serum Nutritional Biomarkers and OSA/OSO

A set of four evaluated studies reported the relationship between nutritional serum biomarkers, namely serum ferritin and 25(OH)D, and OSO [27,28,29,30] presented in Appendix A. One study was derived from a Korean cohort of the Kangbuk Samsung Health Study [27], one from China’s community dwelling women and men [28], and two from the KNHANES databases [29,30]. The total number of evaluated participants was *n* = 39,227, with *n* = 25,427 (64.8%) women. The average age of the participants was 63.1 years (women 62.9 and men 65.4 years). The highest prevalence of OSO was 40.1% and 28.1% in women and men, respectively [29]. The lowest prevalence was 6.4% and 9.4% in women and men, respectively [27] (Appendix A).

Chung et al. [27] examined sex-specific serum ferritin concentration in relation to OSO, reporting that women with the highest ferritin tertiles had the highest OSO prevalence. Additionally, higher serum ferritin concentrations were significantly associated with other adverse body composition impairments in women but not in men. The following three studies [28,29,30] investigated the association of serum 25(OH)D (calcidiol) with OSO phenotype, comparing it with other impairments, namely, osteopenic obesity, sarcopenic obesity, and obesity-only, revealing its supporting role in the pathogenesis of the conditions. Ma Y. et al. [28] examined a large sample of Chinese citizens dividing them into groups based on the body composition impairments and tertiles of serum 25(OH)D. They found that the serum 25(OH)D deficiency was associated with a greater likelihood of having OSO. The results also revealed the independent negative dose-response associations of 25(OH)D with OSO and other impaired body composition components. The two studies [29,30] showed, respectively, that vitamin D deficiency/inadequacy was significantly higher in both women and men with OSO compared to other groups and that higher 25(OH)D was associated with significantly lower odds of having adverse body composition features, especially OSO. Additionally, Kim, Y.M. et al. [29] reported that both women and men in the OSO group engaged in the lowest physical activity compared with those belonging to other groups. Please see Appendix A.

### 3.3. Physical Activity and OSA

In Appendix A, eight interventional studies [31,32,33,34,35,36,37,38] with resistance training are presented. Among these studies, only one included both women and men and a combination of resistance training with aerobic exercise [36]. The total number of evaluated participants was *n* = 182, with the majority being women. The age ranged from 60 to 85 years, and all recruited participants were identified as having OSA/OSO as per inclusion criteria (except Cunha et al. [32]). Three of the studies were respectively conducted on community dwelling individuals in Taiwan [31], Main China [36], and Brazil [32]. The other five papers were published by the same group of researchers from Iran [33,34,35,37,38], where the results from the same participants engaged in the same 12-week interventional resistance training regimen with the elastic band were analyzed but reported different outcome variables affecting women with OSA/OSO (Appendix A).

Overall, each of the papers reported some benefits, including improvements in body fat, muscle mass components (even BMD), some functional performance measures and some biomarkers (see Appendix A). More specifically, Lee et al. and Shen et al. [31,36] reported an increase in BMD, and the former was the only study with a follow-up at 6 months, showing none of the reported benefits were sustained. The study by Cunha et al. [32] is unique as they investigated the response to two resistance training regimens (1-set vs. 3-sets) vs. the control group and found a dose response with higher activity (3 sets induced greater improvement) and both sets induced greater improvement compared to the control group. The authors did not identify OSO as such but created composite Z-scores (derived from average of the muscular strength, skeletal muscle mass (SMM), % body fat, and BMD) and noted a significant improvement after the exercise.

A research group from Iran published several papers with the results derived from the same interventional study with elastic band resistance training in *n* = 63 women (*n* = 32, intervention and *n* = 31, control), all having OSA/OSO as per inclusion criteria [33,34,35,37,38]. The results from their earliest paper revealed a significant increase in handgrip strength, timed chair-rise test, muscle quality, slight improvement in OSO composite Z-score, as well as an increase in estradiol and a decrease in leptin [35]. In one of these papers [33], serum microRNA (miR-133 and miR-206) changes correlated with changes in FRAX scores, serum 25(OH)D, and alkaline phosphatase. Banitalebi et al. [34] reported improved composite cardiometabolic risk factors (e.g., lipid-accumulation product, triglyceride-glucose-BMI index, visceral adiposity index, atherogenic index of plasma, Framingham risk score). Hashemi et al. [37] (from the same group) reported a significant decrease in serum microR-146, total cholesterol, and low-density lipoproteins (LDL) and an increase in high-density lipoproteins (HDL), while Kazemipour et al. [38] reported a significant increase in insulin growth factor (IGF-2), and fibroblast growth factor (FGF-2). However, this intervention did not result in any significant improvement of the body composition components (e.g., BMD, BMI, body fat, skeletal muscle index) or serum biomarkers, such as triglycerides and C-reactive protein (Appendix A).

## 4. Discussion

Based on the results of the studies reviewed, the prevalence of OSA/OSO ranged from 6% to over 90% (see discussion below). Lower intakes of nutrients that emerged as being significantly associated with OSA/OSO (compared to normal or other impaired conditions) were protein, calcium, potassium, and vitamins C and D; all being close to our initial hypotheses and earlier recommendations [7,8,10]. As expected, higher DII scores were associated with OSA or poorer body composition phenotypes, and the opposite was true for the DQI-I and higher low-fat dairy foods and fruit intake. Regarding serum nutritional biomarkers, 25(OH)D was studied the most and surfaced as a beneficial mediator for OSA/OSO and other body composition impairments. A study investigating serum ferritin showed higher values may detrimentally influence body composition (including OSO) but only in women. On the issue of physical activity, several studies evaluated it as a secondary outcome and reported the lowest engagement in recreational activities among participants with OSA/OSO. However, promising results were reported from a few interventional studies with resistance training, showing improvement in various OSA markers, namely, body fat, muscle mass components (even BMD), as well as in some of the functional performance measures and a few of the biomarkers (mostly cardiovascular).

Of note is that all studies were conducted in older individuals (the majority above 60 years of age) when, unfortunately, all components of body composition (bone, muscle, adipose tissues) impairments develop and continue to deteriorate. As it is well recognized, some of the hallmarks of aging are stem cells exhaustion (defaulting to the adipocytes’ commitment differentiation instead of balancing between osteoclasts, myocytes, and adipocytes), telomere shortening, cellular senescence, mitochondrial dysfunction (all leading to chronic inflammation), genomic instability, epigenetic alterations and deregulated nutrient-sensing [39,40], Each of these hallmarks, alone or in combination, ultimately steer to body composition deterioration, in addition to other ailments associated with aging [9,41]. Therefore, investigating OSA/OSO (although an overly complex entity) and taking into account the empirical data from the original studies to support hypotheses can provide the most comprehensive view of various modifiers simultaneously affecting all three body composition components.

Since OSA was defined only in 2014 [1], the studies investigating its relationship with nutrition and physical activity are also relatively new. There are numerous earlier studies measuring/reporting some of the body composition compartments (not necessarily all three) and dietary intake/physical activity. Despite that, we tried to derive OSA from the earlier studies reporting values on bone, muscle, and body fat; none were retrieved to also include nutrition and/or physical activity.

It is disconcerting to note such a high range in OSA/OSO prevalence (about 6–90%; see discussion below). These inconsistencies in literature are due to the lack of universally accepted criteria for OSA identification, as well as the different methods and technologies used to identify it. A long-time agreement exists for the diagnosis of osteopenia/osteoporosis [42] and the revised consensus for sarcopenia diagnosis [43] but not for osteopenic adiposity, sarcopenic adiposity [44], or even adiposity itself [24]. This is unsettling because the inability to determine the prevalence of OSA negatively reflects on its clinical relevance and subsequent treatments and management.

### 4.1. Dietary Intake, Nutritional Status, and OSA

Cvijetic et al. and Keser et al. [16,17] studies are interesting as they are the first ones to measure all three body composition compartments (bone, muscle, and fat) with one bioelectrical impedance (BIA) device (BIA-ACC, BioTekna^®^) to identify OSA. The BIA-ACC instrument is based on bioelectrical impedance, but unlike other BIA devices, it detects the total bone mass (in addition to all soft tissue components and body water). Therefore, its ability to diagnose OSA is quite unique and advantageous due to its simplicity and ease for both patients and researchers. These two studies, although not presenting a strong relationship between OSA and nutritional status and/or specific nutrients, have another merit as they were conducted in several nursing homes and reported the complete assessment of body composition and nutrient intake/status among nursing home residents. Other studies revealed a similar or even higher presence of malnutrition in nursing homes in different countries [45,46]. In view of the advanced age and constrained living conditions of nursing home residents, it is expected that some of them will be malnourished and have a poor body composition status. However, it is surprising that COVID-19 infection in the Cvijetić et al. [16] study participants did not show any impact on OSA prevalence, weight loss, or nutritional status. This is contrary to the just published study in French nursing homes [47], where those infected with COVID-19 experienced about 5% of weight loss. Due to some other limitations of both studies, e.g., cross-sectional nature, a small number of participants in Keser et al. [17], and the inability of BIA-ACC device to distinguish bone mass/quality in different skeletal regions, more studies such as this, where OSA can be diagnosed with one instrument (e.g., BIA or DXA) and conducted in critical populations (like nursing home residents), are warranted.

The results of the 6-month clinical trial of weight loss complemented with low-fat dairy foods or calcium/vitamin D supplements vs. control (placebo group) are clear in showing the improvement of body composition and bone, respectively, despite the relatively moderate weight loss [23]. Numerous studies reported the effects of both dairy foods and calcium/vitamin D supplements on various body composition outcomes. For example, regarding dairy foods and possible mechanisms influencing body composition and weight, several synergistic influential factors have been proposed, including bioactive components (e.g., whey peptides, conjugated linoleic acid, branched chain amino acids), in addition to calcium itself [48]. Regarding the effect of calcium and vitamin D on bone metabolism, as well as weight loss, numerous studies have been published, and the discussion is beyond the scope of this article. For a more in-depth view, please see the discussion in [23]. The prevalence of OSA was not reported in this weight loss study. The participants (although all overweight/obesity by inclusion criteria) only suffered from osteopenia and mild muscle loss and probably only had pre-OSA phenotype. Therefore, these participants, in addition to being younger, were in relatively better shape compared with the participants of other studies, where those with more deteriorated body composition were recruited. There was a considerable attrition rate as well as missing data points in this study—otherwise typical for longitudinal intervention studies—which were counteracted by statistical manipulations. Nevertheless, such kind of studies are valuable in revealing important correlational and causal relations, especially since it was conducted in Caucasian early-postmenopausal women with overweight/obesity—which increased the homogeneity—particularly regarding the bone and body composition that are different among different ethnic and sex groups (e.g., Caucasian vs. African Americans or women vs. men).

Cervo et al. study [25], investigating the relationship between the E-DII and body composition, was included in this review, although OSA was not presented as such, and it was not possible to calculate the prevalence from the presented data. However, the BMD and changes in T-scores were given, along with the number of fractures and fall risks, to account for the bone outcomes. Additionally, handgrip strength was low for both women and men, and so were some other lean/muscle tissue measures, and all participants had overweight/obesity based on the total percent of body fat measured by DXA. Therefore, it could be assumed that most of the participants had pre- or full-OSA phenotypes. Additionally, this was a prospective study with a considerable number of participants followed at 5 and 10 years, and several important variables were measured/assessed. However, the E-DII scores were based on only 19 nutrients, compared to other studies where 27–48 nutrients were used [49]. The E-DII scores also had a relatively narrow range, possibly weakening statistical analyses. The results indicating the association of a higher proinflammatory diet with reduced BMD in both women and men but increased fracture incidence and risk of falls only in men (and decreased in women) are quite intriguing and should be investigated in a more specific way to elucidate these puzzling findings.

The studies conducted in Korea and published by the same group of authors [18,19,20] have several limitations, and some were addressed by the authors. For example, in one study [18], the participants were divided into eight groups, and due to the discrepancy in the distribution of women/men and the number of participants within the groups, both women and men were combined for analysis. The intake of supplements was not considered when assessing the intake of calcium and phosphorus because the focus was on dietary intake (as per the authors’ statement). However, this study had enough participants to classify them into eight distinct groups according to OSA components—the normal group, single component groups, the groups with two OSA components, and the OSA group, thus providing a comparison among each and pointing out to the characteristics of the OSA group. This approach seems to be missing in some other studies due to either a smaller number of participants and/or just non-inclusion of the comparison of OSA with other groups [15,16,17,19,22,23,25]. For example, in another study [19], the authors compared participants with OSO with only those having normal body composition; thus, some possible relationships and comparisons with participants in other groups (e.g., osteopenia, osteopenic obesity or sarcopenia, sarcopenic obesity) in relation to protein intake were not assessed. Additionally, although the authors did not directly report the prevalence of OSO, it could be calculated from the data presented. Accordingly, 649 out of 706 women (91.9%) and 216 out of 645 men (33.4%) were characterized as having OSO [19]. Such high prevalence in women and discrepancy between women and men was not noted in other studies. It could be because women with OSO were significantly older than men with OSO and already had undergone postmenopausal bone and muscle loss and fat accumulation, otherwise not occurring this early and this drastically in men [9]. Additionally, the reported prevalence of OSO in the Choi et al. [19] study was much higher than the prevalence reported in their subsequent study [18] (discussed above), although the same criteria for osteopenia/osteoporosis and obesity were used, and the study was conducted in a similar population (KNHANES 2008–2011 and 2008–2009, respectively). Unfortunately, this discrepancy was not addressed in their subsequent study [18], and it is hard to speculate further. Regarding the last of these three studies [20], it was conducted only in women, and although the DXA scans were utilized for bone and lean tissue assessment, the waist circumference (≥85 cm) as a cutoff for obesity was used, thus making it harder to compare with the studies where DXA or BIA instruments were used. However, the reported prevalence of OSO of 13.7% is more in line with that reported in other studies within community dwelling populations. Although all three mentioned studies [18,19,20] were conducted by the same group of researchers utilizing the KHNAES databases, they revealed quite different prevalence of OSA/OSO (see Appendix A), which was not addressed/discussed by the authors.

The two other studies, Kim et al. and Park et al. [15,22] also analyzed the KNHANES database (2008–2010 and 2009–2011, respectively), investigating the association of the DII and DQI-I, respectively, with OSO prevalence/characteristics. For example, in the Park et al. [22] study, the participants (only females) were classified based on body composition determined by BMI. The limitations of defining overweight/obesity by BMI have been addressed numerous times [2,3,4]. It is hard to justify in this case since DXA was available and used for bone and lean mass assessment. On the upside, the authors used Asia–Pacific guidelines for BMI classifications [50,51] with values for normal weight 18.5 kg/m^2^ ≤ BMI < 23 kg/m^2^, overweight 23 kg/m^2^ ≤ BMI < 25 kg/m^2^, and obesity BMI > 25 kg/m^2^. Despite these limitations, this study analyzed the relationship between OSO and DII using the large KNHANES database, and the results seem to be meaningful as the data may be representative of postmenopausal Korean women. In the Kim et al. [15] study, the association of DQI-I scores with other combinations of abnormal body composition could not be analyzed due to insufficient sample size in other groups. Therefore, the comparison was conducted with healthy Korean young adults. Another drawback was that they defined obesity as the “top 40% of body fat by gender” (as stated in the paper). It is not clear whether they used ≥40% body fat for both women and men, which would be quite high if used for men. Therefore, it is again hard to derive some comparable OSA prevalence value.

The major drawback of the de Franca et al. study [21] was that some study groups comprised a very small number of participants (between six and eleven). Despite this, we decided to include this study in this review because it measured important outcome variables: diet, muscle strength, functional performance, and sedentary lifestyle in relation to body composition and compared OSO participants with those in normal and all others with impaired body composition, although that could be considered as both advantage and disadvantage (considering the small sample sizes in some groups).

Overall, it is encouraging to find many studies addressing the complex entity such as OSA and reporting some promising relations with certain nutrients, including protein, calcium, potassium, vitamins D and C, as well as higher low-fat dairy foods and fruit intake (both reflecting higher quality foods). Weaker relations, yet important, were found with omega-3, fiber, magnesium, phosphorus, and vitamin K. Nevertheless, these findings from the original studies confirmed our earlier hypothetical recommendations in which we emphasized adequate intake of calcium, magnesium, vitamin D, protein, omega-3 fatty acids, and fiber, with restricted energy intake to promote maintenance (or loss) of body weight [13].

### 4.2. Serum Nutritional Biomarkers and OSA/OSO

Appendix A presents studies examining some of the nutritional biomarkers related to body composition and OSO. In view that one of the future hot topics in this area is finding the metabolic profile of OSA/OSO [10], these studies, although limited, are at the forefront of research.

As it is well established, iron overload can promote oxidative stress and cellular membrane damage by generating reactive oxygen species, which is particularly risky for older men and postmenopausal women [52]. Higher iron intake or elevated serum ferritin levels are associated with impaired body composition [53,54]. Some studies reported that iron accumulation in older population could impair bone and muscle metabolism and fat accumulation [55], possibly due to a proinflammatory environment and disturbance of both bone and muscle metabolic pathways. The findings from Chung et al.’s study [27] showed that women with the highest serum ferritin tertile had significantly worse body composition outcomes, including the highest prevalence of OSO, despite that the overall prevalence of OSO in women was lower than that in men (6.4% vs. 9.4%). It is not quite clear why the association of ferritin and OSO (or other impaired body composition components) did not apply to men, despite that the women with OSO were younger than men (66.3 vs. 67.7 years). One of the reasons could have been the classification of participants for both sarcopenia and obesity by using the cutoff values applicable to the Western population. It has been reported that individuals of Asian ethnicity have 3–5% higher body fat for the same BMI values [50]. Therefore, some men might have been missed in the OSO categorization. Nevertheless, this is an interesting and unique study, and the findings could stimulate emphasis on iron intake/overload and body composition in older age and even contribute to developing biomarkers for OSO identification, as suggested by authors [27].

Vitamin D (in its various serum metabolites; calcidiol, calcitriol) is involved in the crosstalk between bone and skeletal muscle by stimulating the production of bone- and muscle-derived factors such as osteocalcin, osteopontin, sclerostin, and myostatin, as well as a vascular endothelial growth factor and insulin growth factor, that all act as endocrine signals between the two tissues. Similarly, low calcidiol (25(OH)D) in obese individuals is still a conundrum as to what the cause and effect are, in view that adipose tissue scavenges much of the circulating calcidiol, but it also provides the source for it [10]. In addition to regulating many metabolic pathways as well as immune response, it is obvious that vitamin D deficiency could have a serious impact on body composition outcomes [56]. While there are numerous studies investigating serum vitamin D inadequacy with various body composition components, so far, only three have been conducted in individuals with OSO [28,29,30]. The results of all three studies revealed the supporting role of calcidiol in offsetting the pathogenesis of the condition. Two studies [29,30] have been derived from the KNHANES in women and men > 50 years and showed that higher serum calcidiol was associated with significantly lower odds of having adverse body composition features, especially OSO, in both sexes. The problem with Kim et al.’s study [30] is that the cutoff for adiposity/obesity was not clearly explained (“in the upper 40% body fat for both sexes”), as addressed above in discussing another article from those authors [15]. This was probably the reason for much higher OSO prevalence and obesity rates in women compared to men (most of them could have been missed). A newer analysis from another group investigating the clinical manifestations associated with OSO [29] using KNHANES confirmed that vitamin D deficiency/inadequacy was significantly higher in both women and men with OSO compared to those with osteopenic obesity, sarcopenic obesity, or obesity-only. Neither of the studies controlled for sun exposure, seasonal variations, or vitamin D supplements, though. These two studies used the same population database (KNHANES IV and V), and the results revealed the same outcomes regarding the relationship between serum vitamin D status and OSO condition. However, the OSO prevalence in the study from 2019 [29] was almost double that of the study from 2017 [30]. The criteria for osteopenia/osteoporosis, sarcopenia, and adiposity in the newer study [29] were more refined or at least better explained. In the earlier study [30], the ALM was not corrected for weight, and the cutoff for obesity was 40% which could have missed many individuals, particularly men (Appendix A). Moreover, although in both studies, the inclusion and exclusion criteria were applied, the study from 2019 [29] had more restricted ones and enrolled a lower number of subjects.

Among the reviewed/published studies, there were no interventional clinical trials which, although smaller in sample size, are controlled and focused, thus could provide more reliable data and cause-and-effect relations. Of the biomarkers, only serum 25(OH)D and ferritin were studied, showing strong relations with OSA, positive and adverse, respectively. Of note is that all the results/analyses from the studies conducted in Korea were extrapolated from the Korean National Databases (KNHANES), with a considerable number of participants providing the epidemiological angle to the issue of OSA/OSO.

### 4.3. Physical Activity and OSA

Recently, a meta-analysis [12] was conducted on four randomized interventional trials also included in this review [31,32,35,36] with resistance/aerobic exercise in OSA/OSO participants, showing improvement in various outcome measures. Interestingly, both Lee et al. and Shen et al. [31,36] reported improvement in BMD, and the former was the only study with a 6-month follow-up revealing, unfortunately, no maintenance of the gained benefits. Both studies had a very small sample size (about *n* = 15 in intervention and control groups), although there were no dropouts, and compliance was good. The improvements in BMD assessed by DXA was noted already after 12 weeks of intervention. However, it is worth noting that the typical change in BMD (without medications or some drastic health deterioration) measured by DXA could be noted only between 5 and 6 months due to the smaller biological/structural changes in BMD in proportion to the instrument’s errors [57]. It is possible that the exercise regimen in the above two studies was very efficient, particularly in the study by Shen et al. [36], where both resistance and aerobic training were employed. Nevertheless, the interpretation of these results should be taken with caution.

The study by Cunha et al. [32] is interesting as the authors were the first ones to utilize a composite Z-score to identify participants with better or worse body composition as an alternative to diagnosing OSO. The composite Z-score was derived from average of the muscular strength, skeletal muscle mass (SMM), % body fat, and BMD and calculated by the formula: (muscular strength Z-score) + (SMM Z-score) + (−1 x body fat Z-score) + (BMD Z-score)/4. The changes induced by the resistance training were evaluated based on the percent change in a composite Z-score. The results also revealed that the three sets of exercise induced better response compared to one set, and significantly so compared to the control group.

All papers by the Iranian group evaluated here described the same 12-week resistance training with an elastic band but analyzed different outcome measures [33,34,35,37,38] (see Appendix A, bottom rows). This was a well-designed and executed intervention in *n* = 63 women with OSA/OSO. While it is commendable to take advantage of the costly studies in human subjects and use the data to examine different outcomes, in this case, the methodology and intervention descriptions were repeated in all papers as well as some of the results (e.g., anthropometry, body composition). Additionally, numerous bone, muscle, and functional performance parameters were measured and reported in each paper, and it was not always clear which ones were used to classify women with OSA/OSO (e.g., composite Z scores or regular cutoffs for each tissue), or functional performance measures for sarcopenia. Nevertheless, these are important findings, and in some of the authors’ more recent papers, the distinctive and novel markers were evaluated, such as serum microRNAs (miR-133 and miR-206), although no change with exercise was found [33]. microRNAs are small, non-coding RNAs involved in the regulation of some physiological and pathological processes, including cell proliferation, apoptosis, and differentiation. It was reported previously that they promoted osteogenic differentiation after osteoporotic fractures [58]. Therefore, the researchers hypothesized that some microRNAs (namely, 133, 206) could be involved in muscle-bone communication and possible transport of positive signals (e.g., induced by exercise) to other tissues. Although a novel idea, the employed exercise regimen did not induce such an effect, and there was no change in these microRNA expressions after the intervention. Another microRNA (miR-146) is known to increase vascular aging by reducing Sirtuin 1 (SIRT1) and activating nuclear factor kappa B (NF-κ B), which both promote vascular smooth muscle cell apoptosis [59]. In their other study [37], the researchers examined the change in miR-146 with resistance training and reported its decreased expression induced by exercise and associated improvements in LDL and HDL. Another paper reported the improvements in some cardiometabolic risk factors, both traditional and composite (e.g., lipid-accumulation product, triglyceride-glucose-BMI index, visceral adiposity index, atherogenic index of plasma, Framingham risk score) with the exercise regimen [34]. Additionally, Kazemipour et al. [38] (from the same group) used a smaller sample (*n* = 48; 26 and 22 in intervention and control groups, respectively) and analyzed some additional markers, including insulin growth factor (IGF-2) and fibroblast growth factor (FGF-2), showing improvement with exercise in both, see Appendix A. (To avoid repeating the same design/intervention for each study in Appendix A, they are listed under Banitalebi et al. [35], with the outcome measures for each noted in the last column).

Overall, the number of studies with exercise intervention is very limited, and yet, these are the only interventional studies with OSA/OSO participants. Additionally, all studies (except one) were conducted on women, and all (except one) employed resistance training, mostly with elastic bands. The studies had a relatively small sample size, and the intervention time was short (12 weeks), which may have impacted the accuracy of the calculations and, ultimately, the interpretation of the results. However, there were no serious side effects, and compliance with exercise was good in all studies, deeming them safe and efficient for improvements of OSA/OSO syndrome in older individuals. Accordingly, we previously recommended a variety of physical activities, including resistance training [13]. The interventions with different exercise modes (aerobic, weight-bearing, and/or comprehensive programs) should be conducted in the future with larger number and more diverse participants and a longer duration of intervention to verify these results, contribute to the overall database, and better understand this complex syndrome.

## 5. Summary and Conclusions

In this scoping review, 23 eligible studies investigating nutritional and physical activity modulators associated with OSA/OSO were evaluated and discussed. OSA syndrome is a complex entity involving the simultaneous impairment of all three body composition tissues (bone, muscle, adipose) with multifaceted etiology and causes, mostly related to the inevitable aging processes. Despite that the OSA syndrome has been identified relatively recently, the research is picking up, and numerous studies have been conducted, although not all had the elements required for inclusion in this review. This was particularly a case for the older studies where each of the body composition components was studied separately or, at the most, as a combination of two impaired conditions (e.g., sarcopenic obesity) [1]. Additionally, in some instances where all three body composition components were included/studied, it was not always possible to connect them simultaneously to derive OSA/OSO and relate to the independent/exposure variables, in this case, nutrition, serum nutritional biomarkers, and physical activity. Nevertheless, this review revealed some important findings contributing to a better understanding of the OSA syndrome in older individuals (particularly women), providing empirical data for the theoretical hypotheses.

The most important findings regarding the nutrients/dietary intake and OSA revealed that higher protein, calcium, potassium, and vitamins D and C intakes, in addition to consuming enough fruit and low-fat dairy foods—comprising antioxidative nutrients (fruits) and food of high nutritional quality (both)—emerged as most beneficial. Some other nutrients, although to a lesser extent, appeared important and in line with our previous recommendations [7,8,10,13]. These include omega-3 polyunsaturated fatty acids, fiber, magnesium, and vitamin K. No interventional or clinical trials were conducted to investigate serum biomarkers and determine the metabolic profile of OSA, which currently is a much sought and needed research. Among four studies, three showed a beneficial association between higher serum 25(OH)D and OSA, and one (quite new and surprising) showed higher serum ferritin levels being associated with worsened OSA outcomes. Several studies reported the benefits of recreational activities and a less sedentary lifestyle on OSA; however, the promising ones were the interventional studies with resistance training, all conducted in women with OSA. Among the most beneficial outcomes were the improvements in body composition (body fat, muscle), most of the functional performance measures, and some of the biomarkers (mostly cardiovascular). Therefore, based on the reviewed studies, nutrient-dense food with plenty of protein, calcium, and vitamins D and C, along with an active lifestyle and possibly some organized and safe exercise, such as resistance training, would be beneficial to maintain good body composition and manage OSA. Additionally, multi-component interventions targeting each distinct component of OSA are likely to benefit older adults with overweight/obesity and poor bone and muscle health. We and others have proposed lifestyle approaches for musculoskeletal health in these individuals, which incorporate a combination of dietary/energy restriction, with adequate intakes of protein, calcium, and vitamin D, and also progressive resistance training and weight-bearing impact exercise to offset and potentially reverse weight loss-associated declines in bone and muscle mass [6,7,13]. This type of diet and exercise regimen is beneficial for overall health, especially in the aging population [60,61].

## 6. Existing Problems and Recommendations for Future Studies

The operational definitions and diagnostic criteria are already available for osteopenia/osteoporosis [42] and sarcopenia [43] and, to some extent, for sarcopenic obesity. For the latter, a consensus statement was just revised and published last year with the updated definition of sarcopenic obesity, its diagnostic criteria to include both gold standards and other acceptable alternates, as well as methodologies to be used with the related cutoffs [44]. Unfortunately, substantial incongruity exists on standardized cut-points for body fat percentage to define obesity and even more so for visceral adiposity. This is due to numerous reasons (e.g., adipose tissue heterogeneity and demographic factors such are age and ethnicity), as was discussed recently [2]. Additionally, based on the OSA definition, it is not just the amount of fat tissue (as overt overweight/obesity), but also the kind of fat (visceral, subcutaneous) and infiltrated fat into the bone and muscle which creates the problems as the latter two are not easily measured and detected [2]. As we proposed earlier [7], some other measures, including waist circumference, may be used in place of body fat percentage or as an additional measure to identify adults with visceral adiposity, especially if more advanced technology (iDXA, CT) is not available. Therefore, identifying OSA still depends on each researcher’s abilities, institutional infrastructure, and available equipment.

It is clear that the lack of universally established diagnostic criteria for OSA/OSO hinders patients’ identification, worldwide determination of its prevalence, and assessment of related outcomes, as well as the creation of any public health policies. Even more importantly, the inability to properly identify OSA/OSO patients impedes the development of its prevention and treatment strategies. This very review exposed some of the marked discrepancies, namely, different diagnostic approaches which originate from different criteria used for classifications of obesity and sarcopenia, methods to assess functional abilities, as well as differently applied reference values for the variables examined and statistical stratification. Once when the consensus on the definition and diagnostic criteria is established, the evaluation of patients for OSA/OSO should become a part of the routine clinical practice.

Under the broader umbrella and based on the evidence presented, it is obvious that large observational, longitudinal, and interventional (in the form of randomized controlled trials) studies are needed to further elucidate the distinct as well as the synergistic effects of nutritional and/or exercise roles in individuals with OSA. The intervention should have a specific focus on the OSA physical phenotype and on the metabolic and inflammatory biomarkers. This will enable the clarification of the role of specific nutrients and/or exercise regimens in the pathogenesis, management, and treatment of OSA. In this context, the prospective studies and the secondary analysis of existing datasets (the cheapest) to study the possible predictors and clinical impact would be of great value.

Besides examining nutrition and physical activity in relation to OSA, a lot more is needed in this area, including, but not limited to:The participants with OSA should be compared with those having osteopenia/osteoporosis, osteopenic adiposity, sarcopenia, sarcopenic adiposity, osteopenic sarcopenia, adiposity-alone or normal-body composition parameters (see Figure 1 for the combination of conditions). This will provide a clearer picture of OSA itself and all the differences between other body composition impairments.Individuals of different sex (as of now, women are studied more frequently than men), ages, and race/ethnicities (e.g., there are no studies in African Americans), as well as critical populations (like nursing home residents), are needed to better define the diagnostic criteria and to elucidate OSA. While majority of the studies have been conducted in older population, equally important would be the studies in younger individuals, as the earlier work identified prevalent OSA phenotype in healthy, young, obese individuals [62].The potential breakthrough could be the development of biomarkers for each tissue which in combination may indicate the existing impairments and presence of OSA. A pilot study showed increased levels of serum sclerostin (bone resorption marker), skeletal muscle troponin (muscle breakdown marker), and inferior lipid profile and increased leptin in women with OSA compared to their counterparts with only one or two impaired body composition components [63]. However, more refinement is necessary, and the series of omics will need to be determined to serve as potential biomarkers.Likewise, in view of the swift technological advances, such as genomic sequencing and molecularly targeted drug exploitation, the concept of precision medicine can be used to demarcate OSA using multiple data sources from genomics to digital health metrics to artificial intelligence in order to facilitate an individualized yet “evidence-based” decisions regarding diagnostic and therapeutic approaches. In this way, therapeutics can be centered toward patients based on their molecular presentation rather than grouping them into broad categories with a “one-size-fits-all” approach.

## Figures and Tables

**Figure 1 nutrients-15-01619-f001:**
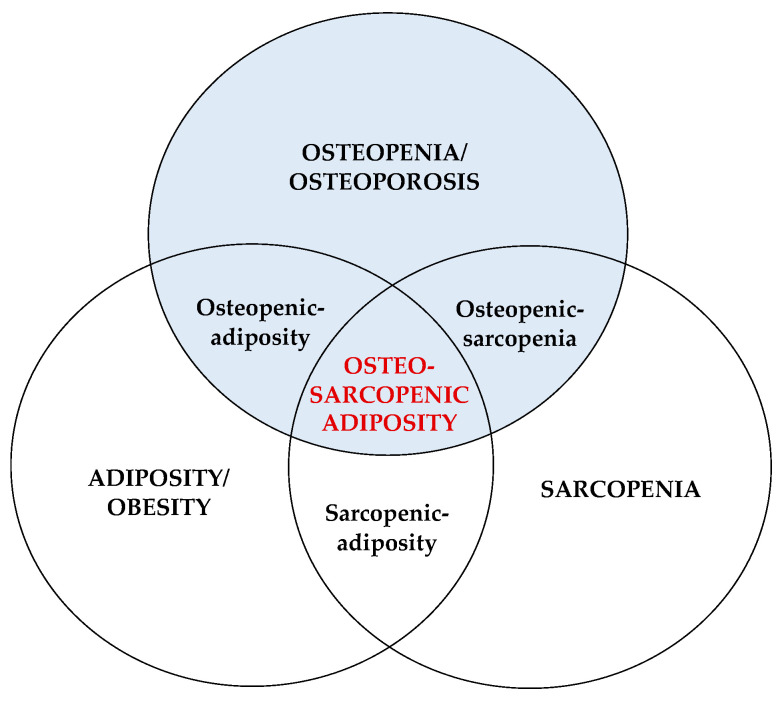
Osteosarcopenic adiposity/obesity and its components. Adapted from Ilich et al. [1].

**Figure 2 nutrients-15-01619-f002:**
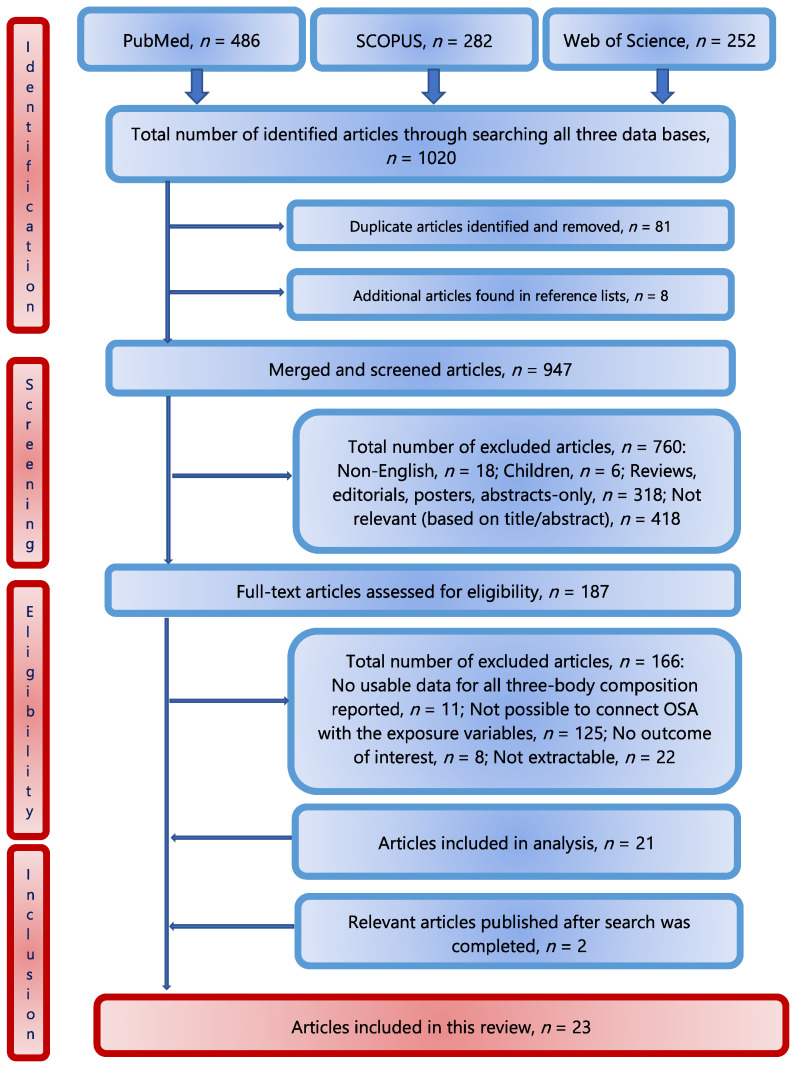
Flowchart (PRISMA) of the study selection process.

## Data Availability

Not applicable.

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
