# Peer review of "Nutrition and Physical Activity as Modulators of Osteosarcopenic Adiposity: A Scoping Review and Recommendations for Future Research"

_nutrients, 2023, doi:10.3390/nu15071619_

Round 1
Reviewer 1 Report
Dear authors,
I found the content of your manuscript interesting and valuable.
However, I have several remarks:
1. Figure 1. Could you make again this figure using the same fonts and font sizes?
2. Figure 2. Please correct the fonts in the red squares.
3. Table 1. Could you make it more user friendly.
4. Tables 2 and 3: Check the fonts of the description of the table. Please follow MDPI instructions to authors.
5. Table 3. Could you make it more user friendly.
Author Response
We want to thank the editors and the reviewers for their effort and valuable suggestions. We addressed them in the paper (highlighted in yellow) and noted below, point by point. The comments made great improvement to our manuscript, and we hope that the manuscript in current form will be acceptable for publishing in Nutrients.
REVIEWER 1:
I found the content of your manuscript interesting and valuable.
Thank you for this encouraging comment.
However, I have several remarks:
- Figure 1. Could you make again this figure using the same fonts and font sizes?
The fonts were all adjusted to the same size. The capital-case letters demarcate the main body composition compartments and osteosarcopenic adiposity (the focus of this paper), while the lower-case letters demarcate the derived body composition compartments. The Figure 1 now looks better, thank you.
- Figure 2. Please correct the fonts in the red squares.
The fonts in all squares have been adjusted to the Paladino font (as per journal guidelines), and those in red squares are made in the same style (lower-case). And thank you again, the Figure 2 looks better now.
- Table 1. Could you make it more user friendly.
- Tables 2 and 3: Check the fonts of the description of the table. Please follow MDPI instructions to authors.
- Table 3. Could you make it more user friendly.
We appreciate the reviewer’s suggestion and fully understand this technical weakness. We were aware that the tables were not up to the publishing format. The fonts in all tables have now been fixed as per the MDPI guidelines and the tables were adjusted to be more “user friendly”. However, the tables contain a lot of a very important material from each of the evaluated article and could not be substantially shortened. Therefore, we submitted the tables as part of the supplementary material.
Reviewer 2 Report
It is a very interesting subject, and it fits in the scope of Nutrients “Nutrition and Physical Activity as Modulators of Osteosarcopenic Adiposity: A Scoping Review and Recommendations for Future Research”
In this paper, the authors attempt to provide a scoping review of the literature on the relationship were to evaluate human studies addressing dietary intake/nutritional status, and the quantity/types of physical activity related to OSA.
Introduction
I am not sure if the conceptual framework that allowed authors to perform this investigation is well described.
I propose renaming the condition Osteosarcopenic Obesity instead of Osteosarcopenic Adiposity.
Methods
Only research involving middle-aged and/or older people were permitted by the authors (generally the most affected).
The tables appear confused and are difficult to read.
The fonts and sizes don't seem to match the Nutrients template.
Results
On this portion, I have no worries.
Discussion
I believe that a more thorough explanation of the findings considering earlier
research could enhance this section.
Author Response
We want to thank the editors and the reviewers for their effort and valuable suggestions. We addressed them in the paper (highlighted in yellow) and noted below, point by point. The comments made great improvement to our manuscript, and we hope that the manuscript in current form will be acceptable for publishing in Nutrients.
REVIEWER 2:
It is a very interesting subject, and it fits in the scope of Nutrients “Nutrition and Physical Activity as Modulators of Osteosarcopenic Adiposity: A Scoping Review and Recommendations for Future Research”
We appreciate the reviewer’s positive comment.
In this paper, the authors attempt to provide a scoping review of the literature on the relationship were to evaluate human studies addressing dietary intake/nutritional status, and the quantity/types of physical activity related to OSA.
Introduction
I am not sure if the conceptual framework that allowed authors to perform this investigation is well described.
Thank you for this comment and yes, you are right. We did not justify enough the need for this particular literature search to evaluate the nutritional and physical activity associations with osteosarcopenic adiposity. We added a paragraph in the Introduction.
I propose renaming the condition Osteosarcopenic Obesity instead of Osteosarcopenic Adiposity.
We appreciate this comment, however, the name osteosarcopenic obesity/adiposity has a history. Originally, the syndrome was coined as osteosarcopenic obesity in 2014 (Ilich 2014). Subsequently, realizing the complexity and heterogeneity of adipose tissue (subcutaneous, visceral, ectopic) and the typical meaning of obesity (overt overweight/obesity mostly indicated by high BMI), we changed the name into osteosarcopenic adiposity (Ilich 2020). This term reflects better the condition, which demarcates not just the excess of body fat but also the infiltrated and ectopic fat into bones and muscles. The obesity has also been addressed as the Adiposity-Based-Chronic-Disease (ABCD) by the American Association of Clinical Endocrinologists and American College of Endocrinology in their position paper (Mechanick, 2017). This issue was reinforced and better explained in the Introduction.
Mechanick JI, Hurley DL, Garvey WT. ADIPOSITY-BASED CHRONIC DISEASE AS A NEW DIAGNOSTIC TERM: THE AMERICAN ASSOCIATION OF CLINICAL ENDOCRINOLOGISTS AND AMERICAN COLLEGE OF ENDOCRINOLOGY POSITION STATEMENT. Endocr Pract. 2017 Mar;23(3):372-378. doi: 10.4158/EP161688.PS. Epub 2016 Dec 14. PMID: 27967229.
Methods
Only research involving middle-aged and/or older people were permitted by the authors (generally the most affected).
The tables appear confused and are difficult to read.
The fonts and sizes don't seem to match the Nutrients template.
We apologize for this mishap and inconvenience. We adjusted the fonts and submitted the tables in the supplementary material as they were too big for the manuscript text in the template.
Results
On this portion, I have no worries.
Discussion
I believe that a more thorough explanation of the findings considering earlier
research could enhance this section.
The osteosarcopenic obesity/adiposity syndrome was identified in 2014. The original research on the topic started only after that time. We searched all papers from their inception to present time to find those in which all three body composition compartments were reported (in order to identify osteosarcopenic obesity/adiposity) and related to the nutrition or physical activity. The earlier studies just did not have all the components necessary to inclusion in this review. We added this in Discussion and we appreciate your pointing this out. We also discussed the findings of this review with our previous recommendations for both nutrition and physical activity regarding the osteosarcopenic adiposity (as highlighted).
Round 2
Reviewer 2 Report
I congratulate the authors for their work and I accept in present form.